# Antibiogram of uropathogens and associated risk factors among asymptomatic female college students in Dessie town, Northeast Ethiopia

**Berhanu Kebede Reda** [1]*, **Genet Molla**[2], **Alemu Gedefie**[2], **Daniel Gebretsadik** [2], **Mihret Tilahun** [2], **Melaku Ashagrie Belete**[2], **Agumas Shibabaw**[2]

1 Department of Medical Laboratory Science, College of Medicine and Health Science, Samara University, Samara, Ethiopia, 2 Department of Medical Laboratory Science, College of Medicine and Health Sciences, Wollo University, Dessie, Ethiopia

* birekebede666@gmail.com

## Abstract

### Background

Asymptomatic urinary tract infection (asymptomatic bacteriuria and asymptomatic candiduria) may not be routinely detected in sexually active non-pregnant female population at the initial and reversible stages. This is mainly due to the fact that most women may not feel compelled to seek medical attention.

### Objectives

The aim of this study was to determine the prevalence, and factors associated with urinary tract infection (UTI), and antibiogram of the uropathogen isolates among asymptomatic female college students.

### Methods

An institutional-based cross-sectional study was conducted at selected colleges in Dessie from January 2021–March 2021. A total of 422 reproductive age (15 to 49 years) non-pregnant female students were included. Socio-demographic and clinical characteristics data were collected using structured questionnaires. Ten mLs of freshly voided mid-stream urine specimen was collected, transported and processed according to the standard operating procedures. Data were coded and entered for statistical analysis using SPSS version 22.0. Descriptive statistics, bivariate and multivariate logistic regression analysis were performed and p-values <0.05 with the corresponding 95% confidence interval (CI) were considered statistically significant.

### Result

The overall prevalence of UTI was 24.6%. The prevalence of asymptomatic UTI bacteriuria and candiduria was 57 (13.5%) and 47 (11.1%), respectively. The predominant

**Data Availability Statement:** All available data were included in the paper and its Supporting information files.

**Funding:** no. The author(s) received no specific funding for this work.

**Competing interests:** no. The authors have declared that no competing interests exist.

**Abbreviations:** AHBSC, Alkan Health and Business Science College; AOR, adjusted odds ratio; AST, Antimicrobial Susceptibility Testing; CFU, Colony Forming Unit; CI, confidence interval; CLED, Cystine Lactose Electrolyte-deficient Agar; CLSI, Clinical and Laboratory Standards Institute; COR, crude odds ratio; DHSC, Dessie Health Science College; DM, Diabetic Mellitus; ETB, Ethiopian birr; I, Intermediate; IUD, Intra uterine device; MC, Memheran College; MDR, Multi Drug Resistance; R, Resistant; S, Sensitive; SDA, Sabouraud dextrose agar; SOPs, Standard Operating Procedures; SPSS, Statistical Package for the Social Sciences; STDs, Sexually Transmitted Diseases; TCOM, Tropical College of Medicine; UTI, Urinary Tract Infection.

uropathogens were *Staphylococcus saprophyticus* 24 (23.1%), followed by *Candida tropicalis* 23 (22.1%), *Candida albican* 10 (9.6%), *Candida krusei* 9 (8.7%) and *Escherichia coli* 8 (7.7%). Gram negative bacterial isolates showed a higher level of resistance to amoxicillin-clavulanic acid 24 (92.3%). Gram positive bacterial uropathogens showed high level of resistance to penicillin 28 (96.6%) and trimethoprim-sulfamethoxazole 23 (79.3%). Gram positive bacterial isolates were sensitive to norfloxacin, clindamycin, and ciprofloxacin, accounting for 24 (82.7%), 20 (69.0%), and 19 (65.5%), respectively. Multidrug resistance was seen in 50 (87.7%) of bacterial uropathogens. Factors identified for acquisition of UTI were frequency of sexual intercourse ($\geq$3 per week) (AOR = 7.91, 95% CI: (2.92, 21.42), and genital area washing habit (during defecation (AOR = 5.91, 95%CI: (1.86, 18.81) and every morning (AOR = 6.13, 95%CI: (1.60, 23.45)).

## Conclusion

A significant prevalence of uropathogens, and high resistance of bacterial isolates to the commonly prescribed drugs were detected. Therefore, routine UTI screening, regular health education on the risk of asymptomatic infectious diseases for reproductive age group females, and antimicrobial susceptibility testing should be practiced to avoid the progression of an asymptomatic infection into a symptomatic UTI.

## Introduction

Urinary Tract Infection (UTI) is an infection caused by microorganisms affecting different parts of the urinary system, including the urethra, bladder, ureters, and kidneys [1]. Asymptomatic uropathogens are frequently encountered and contribute significantly to UTIs, often leading to unnecessary treatments [2]. This condition affects millions of people worldwide and can progress to severe symptomatic infections, particularly in developing countries where the healthcare coverage is comparably low [3]. Globally, it is estimated that around 150 million people experience bacteriuria annually, either symptomatically or asymptomatically [4]. It has been shown that asymptomatic bacteriuria is more prominent in women compared to men [2].

The most common causative agent for UTI are bacteria including *Escherichia coli*, *Proteus mirabilis*, *Klebsiella* species, *Pseudomonas aeruginosa*, *Enterobacter* species, *Enterococci*, *Citrobacter*, *Staphylococcus aureus*, *Staphylococcus saprophyticus*, *Streptococci* species and other fungi agents such as *candida* species also contribute for development of UTI [5]. Women are three times more likely to be affected by UTI than men, because of shorter urethra, nature of sexual activity, pregnancy, easy contamination of the urinary tract with faecal flora and hormonal changes that occur quickly [6].

Asymptomatic infections do not show any sign and symptoms of infection, whereas symptomatic infections do. Asymptomatic UTI, also known as asymptomatic bacteriuria (ASB) or asymptomatic urinary candiduria is defined as the presence of significant bacteria ($\geq 10^5$ CFU/mL) or a count of $\geq 10^5$ CFU/mL candida species in an individual's urine without signs and symptoms of UTI [7, 8]. Age, sex, sexual activity, and the existence of genitourinary anomalies all play a great role in the occurrence of UTI [9].

Candida species have been identified as one of the most common causal agents of UTI caused by fungi. Immunosuppression, extremes of age, diabetes mellitus, structural abnormalities of the urinary tract, broad-spectrum antibiotic use, admission to an intensive care unit

(ICU), females of the reproductive age, indwelling catheters, and abdominal surgeries are common risk factors for candiduria [10]. Affected individuals with candiduria are asymptomatic, and yeasts are frequently isolated in urine culture [11].

Asymptomatic uropathogens causing UTI among sexually active non-pregnant females may not be detected in the early and reversible stages in most cases [12]. This is due to the fact that most women may not feel compelled to seek medical attention. Significant asymptomatic bacteriuria has been proposed to indicate an early clinical history of UTI, which could lead to acute infection, chronic infection, or even death due to kidney failure [13]. Even if the infection appears to be minor at first, the patient may experience symptoms as the infection progresses [14].

UTI is one of the most common community-acquired infection caused by uropathogens and is a common reason for seeking medical attention in the community. Rapidly increasing antimicrobial resistance of uropathogens results limited treatment options. Therefore, knowledge of the uropathogens and their antibiotic susceptibility pattern is important for better treatment of UTI [15].

Few studies have been conducted in Africa regarding the magnitude and drug susceptibility pattern of asymptomatic uropathogens among reproductive age non pregnant female students showing the prevalence of UTI in Ghana (9.6%) and Nigeria (13.8%) [16, 17]. On the other hand, the prevalence of candiduria in HIV positive women attending at a hospital in Cameroon and asymptomatic candiduria in diabetic patients attending in Nigeria was (21%) and (40.9%), respectively [18, 19]. However, most of the study was occasioned by the apparent lack of data [4].

The majority of studies conducted in Ethiopia have primarily focused on pregnant women, revealing a range of prevalence rates for bacteriuria (8.5% to 18.8%) across various regions of the country [20–26]. Other studies on asymptomatic bacteriuria focus on the risk groups such as diabetic patients (16.7%), HIV patients (18%), pediatric patients (15.9%), antiretroviral therapy users (11.2%) [27–30]. Similarly, studies on asymptomatic candiduria focused primarily on risk factors, such as female asymptomatic diabetic patients (36.8%) [31].

The majority of studies conducted in Ethiopia have primarily focused on pregnant women and other high-risk groups, such as diabetic patients, HIV patients, and pediatric patients [27–31]. However, there is a notable gap in documented information regarding the population of non-pregnant women of reproductive age, despite this group being at a higher risk of UTI. To address this gap, the present study was designed to investigate the profile of uropathogens, factors associated with asymptomatic UTIs, and antibiogram of bacterial isolates among female college students in Dessie town, Northeast Ethiopia.

## Material and methods

### Study area, design and period

The study was conducted at selected colleges in Dessie town, Northeast Ethiopia from January 2021–March 2021. Specifically, the study was conducted in Dessie Health Science College (DHSC), Memheran College (MC), Tropical College of Medicine (TCOM), and Alkan Business and Health Science College (ABHSC). Dessie Health Science College and MC are government-owned colleges, while TCOM and ABHSC are privately owned colleges. The study deployed an institutional-based cross-sectional study design.

### Sample size determination and sampling technique

The sample size was determined using single proportion formula by considering 50% estimated prevalence (p), 0.05 margin of error (d), and 95% level of confidence (z = 1.96).

 

Therefore, a total of 422 non-pregnant asymptomatic female college students were included using a systematic random sampling technique after proportional allocation was used to determine the number of study participants from the four colleges based on the population size. The colleges included in the study were selected using a simple random sampling technique from a total of ten colleges in the town. A systematic random sampling technique were used to enrol female college students of all batches (from 1st year up to 4th year) attending class at the selected colleges during the study period fulfilling the entry criteria. The distribution of the study participants is calculated using proportionate allocation formula; and accordingly, 132 study participants from Tropical College of Medicine, 125 study participants from Dessie Health Science College, 109 participants from Alkan Health and Business Science College and 56 participants from Memheran College were recruited. Based on the proportional allocation after the alphabetical list of female students in each batch is obtained from the registrar office of each college, the list was served as sampling frame and the sampling interval (k) was calculated discretely for each college by dividing the number of female students in the batch to the allocated sample size. Depending on the value of the sampling interval, the first study participant was selected randomly, and then every kth female student were selected to participate in the study.

Non-pregnant female college students without symptoms of UTI who were willing to participate in the study were included. While female college students who were pregnant, had UTI symptoms, or refused to give consent to participate in the study were excluded. Moreover, students who were taking antibacterial or antifungal drugs during or within the past 2 weeks of data collection were excluded.

## Data and specimen collection

A structured self-administered questionnaire was used to collect demographic and clinical characteristics data of study participants. The questionnaire was close-ended, and had 3 sections including socio-demographic characteristics (age, residence, student batch, marital status, monthly family income), hygiene-related habits and clinical risk factors (histories of UTI, catheterization, DM, HIV/AIDS, STDs, hospitalization, antibiotic use, use of contraceptives, genitourinary abnormality, sexual activity and genital area washing habit). About 10 mLs freshly voided clean-catch midstream urine specimen was collected using pre-labeled (identification code, time, date), wide mouth, leak-proof, sterile, screw-capped plastic container (FL Medical, Italy) by study participants after appropriate specimen collection instructions were given.

## Specimen transportation

The collected specimens were transported to Wollo University Microbiology Laboratory using cold box within two hours for processing after confirming their non-pregnancy status by using human chorionic gonadotropin (HCG) card pregnancy test immediately after collection. In case of unavoidable delay, very few specimens were kept refrigerated at 4°C until being processed. Immediate inoculation had been performed for the rest of the specimens on arrival to the laboratory.

## Uropathogen culture, isolation and identification

Urine sample (0.001mL) was inoculated onto Cystine Lactose Electrolyte Deficient medium (CLED) (Oxoid Ltd, UK) using calibrated wire loop. Inoculated plates were incubated overnight in aerobic atmosphere at 37°C for 24 hours. Colonies were counted to check the presence of significant growth. Colony counts yielding bacterial growth of $\geq 10^5$ CFU/mL of urine was

regarded as significant bacteriuria (SB); but specimens that produce $<10^5$ CFU/mL were considered insignificant or due to contamination [32].

Based on their gram staining reaction, colonies from CLED were sub cultured onto MacConkey agar (Oxoid, Ltd), blood agar plates and Mannitol salt agar (Hi Media™), then incubated at 37˚C for 24 hours. Identification of bacterial species was done using colony characteristics, gram staining reaction and standard biochemical tests. The gram-negative bacteria were identified by indole production, motility test, citrate utilization, urease test, $H_2S$ production in Kligler's Iron agar (KIA) and carbohydrate utilization tests. The gram-positive bacteria were identified using catalase, coagulase tests and novobiocin susceptibility test [32].

For fungal identification, a loopful (0.001 mL) of well-mixed un-centrifuged urine was streak onto the surface of Sabouraud dextrose agar (SDA) [32]. The plates were incubated aerobically at 37˚C for 24–48 hours. A count of $\geq 10^5$ CFU/ml was considered an indicator of asymptomatic urinary candiduria [8]. The significant fungal isolates were sub-cultured onto CHROM agar TM Candida media (chrome agar, Paris, France). All Fungal isolates were identified with CHROM agar TM Candida Media to confirm the presence of the five *Candida species* (*C. albicans*, *C. tropicalis*, *C. krusei*, *C. glabrta* and *C. auris*) [33]. Furthermore, all uropathogen isolates were identified phenotypically using standard clinical laboratory methods [32].

**Antimicrobial susceptibility testing.**   The antimicrobial susceptibility testing (AST) of all identified bacterial isolates were performed using the Kirby-Bauer disc diffusion method on Muller-Hinton agar (MHA) according to the Clinical and Laboratory Standards Institute (CLSI) guideline [34]. A loop full of bacteria was taken from a pure culture colony and transferred to a tube containing 5 mLs of normal saline and mixed gently until it forms a homogenous suspension. The turbidity of the suspension was adjusted to the turbidity of 0.5 McFarland in order to standardize the inoculum size. A sterile cotton swab was dipped, rotated across the wall of the tube to avoid excess fluid and was evenly inoculated on MHA (Conda ltd, USA) and then the antibiotic discs were placed on MHA plates.

The following antimicrobials were used based on the CLSI recommendations and local commonly prescribed drugs for UTI treatments in the study area, including clindamycin (CL, 10μg), penicillin (PEN, 10μg), chloramphenicol, (CAF, 30μg), ciprofloxacin (CIP, 5μg), Tetracycline (TTC, 30μg), trimethoprim sulfamethoxazole (SXT, 1.25/23.75μg), nitrofurantoin (F, 300μg), and norfloxacin (NOR, 10μg) for Gram positives; on the other hand ciprofloxacin (CIP, 5μg), tetracycline (TTC, 30μg), trimethoprim sulfamethoxazole (SXT, 1.25/23.75μg), nitrofurantoin (F, 300μg), norfloxacin (NOR, 10μg), ceftriaxone (CRO, 30μg), amoxicillin-clavulanic acid (AMC, 20/10μg), cefotaxime (CTX, 30μg), ceftazidime (CAZ, 30μg), ampicillin (AMP, 10μg), amikacin (AMK, 30μg) and gentamicin (GN, 10μg) were used for Gram negative bacteria.

The plates were incubated overnight at 37˚C for 24 hours. Diameters of the zone of inhibition around the discs were measured using a digital caliper. The AST result interpretation was based on the CLSI [34] criteria as sensitive, intermediate and resistant.

## Quality assurance

All quality control tests were performed before, during, and after data collection to ensure that the data was of high quality and reliability. All questions in the structured questionnaire were written in a clear and explicit manner and then translated into the local language (Amharic). An hour training was given to data collectors on data collection. The completeness of the questionnaires was checked by principal investigator. The questionnaire was pretested on 5% of the sample size (22 asymptomatic female students) from Mankul College which is located in

Dessie city, Northeast Ethiopia, and necessary modifications were made to the data collection tool accordingly, particularly to improve the clarity of questions. Moreover, all laboratory assays were done by maintaining the quality control procedures. The performance of Gram staining chemical was checked by colonies of *S. aureus* (ATCC 25923) and *E. coli* (ATCC 25922). Standard Operating Procedures (SOPs) were strictly followed verifying that media meet expiration date and quality control parameters per CLSI guideline. Media sterility was checked by incubating 5% of the prepared media (batch) overnight and looking for growth after 24hrs. For biochemical media, one organism that was produced a positive reaction and one organism that negative reaction were applied. Visual inspections of holes, uneven filling, and haemolysis, signs of freezing, bubbles and corrosion in media or plastic Petri dishes was conducted. standardized inoculum density of bacterial suspension was prepared, saline turbidity standard equivalent to a 0.5 McFarland standard was used to control the quality of antimicrobial susceptibility test. Standard reference strain of *S. aureus* (ATCC-25923), *E. coli* (ATCC-25922), *K. pneumoniae* (ATCC 700603) and *C. albicans* (ATCC 90028) were used as quality control strains throughout the study. Moreover, the quality of MHA was checked using *Enterococcus faecalis* isolates and a corresponding trimethoprim-sulfamethazine antimicrobial disk in accordance with CLSI standards.

## Statistical analysis

The collected data were entered and analyzed using Statistical Package for Social Sciences (SPSS) version 22.0 (IBM, USA). Descriptive statistics were computed, and data were presented using frequency and percentages. A total of 15 independent variables (socio-demographic characteristics, hygiene-related habits, and clinical factors) were considered during the bivariate analysis of associated factors for UTI. Binary logistic regression was used to show the relationship between each variable and the dependent variable (significant uropathogen). Variables that showed significance at p-values of $\leq 0.25$ during univariate analysis (only 8 independent variables) were selected for further multivariable analysis. Moreover, a multivariate analysis was employed to avoid confounding variables and identify factors that independently influence the occurrence of the dependent variable. Adjusted odds ratios (AOR) with 95% confidence intervals (CIs) were used as indicators of the strength of association. P-value $< 0.05$ with 95% confidence interval was considered statistically significant.

## Ethical approval and consent to participate

The ethical approval was obtained from College of Medicine and Health Science Research Ethics Review Committee (RERC), Wollo University with reference number: CMHS-210/13/21. Official cooperation and permission letters were obtained from selected colleges in Dessie town. Moreover, prior to commencing the study, a written informed consent was obtained from each study participant. Subject confidentiality and any special data security requirements were maintained and assured. Results of the laboratory examinations that have a direct benefit in the health of the study participants were linked to a health facility and the participants get their results and treatment duly as required. Moreover, this study was conducted in accordance with the Declaration of Helsinki.

## Operational definitions

**Multiple drug resistance (MDR):** non-susceptibility to 1 or more antimicrobial agent in 3 or more antimicrobial categories.

# Results

## Socio-demographic characteristics

A total of 422 female college students were investigated for asymptomatic UTI. The age of study participants ranged from 18 to 38 years, with a median age of 21 years and an interquartile range (IQR) of 20 to 23 years. About half of the study participants 209 (49.4%) were in the age group of 21 to 25 years. Among the participants, their student batch varied from second year to fourth year, with the second-year batch comprising the largest proportion of study participants 177 (41.9%) (Table 1).

## Prevalence of urinary tract infection

The overall prevalence of UTI was 104 (24.6%). The prevalence of significant bacteriuria and candiduria was 57 (13.5%) and 47 (11.1%), respectively.

## Associated risk factors of UTI

In this study, 15 independent variables were considered during the bivariate analysis of risk factors for bacterial and candidal UTI. Of these, only 8 independent variables show significance at

**Table 1. Socio-demographic characteristics of the study participants in Dessie town, Northeast Ethiopia, January 2021–March 2021.**

| Variables | Number (%) |
| --- | --- |
| **Age (years)** | |
| ≤20 | 172 (40.7) |
| 21–25 | 209 (49.4) |
| 26–30 | 25 (5.9) |
| >30 | 16 (3.8) |
| **Residence** | |
| Urban | 388 (91.9) |
| Rural | 34 (8.1) |
| **Colleges** | |
| TCOM | 132 (31.3) |
| DHSC | 125 (29.6) |
| AHBSC | 109 (25.8) |
| MC | 56 (13.2) |
| **Student batch** | |
| Second year | 177 (41.9) |
| Third year | 161 (38.2) |
| Fourth year | 84 (19.9) |
| **Marital status** | |
| Single | 335 (79.4) |
| Married | 69 (16.4) |
| Divorced | 18 (4.3) |
| **Monthly family income (ETB)** | |
| ≤1000 | 37 (8.8) |
| 1001–2000 | 72 (17.1) |
| 2001–3000 | 28 (6.6) |
| 3001–4000 | 52 (12.3) |
| >4000 | 233 (55.2) |

p-values of ≤ 0.25 in the bivariate analysis, and were selected for multivariable analysis. In the bivariate analysis, contraceptive use (COR = 2.279, 95%CI: (1.112, 4.672), P = 0.024), genital area washing (during defecation (COR = 3.452, 95%CI: (1.382, 8.622), P = 0.008) and every morning (COR = 3.366, 95%CI: (1.117, 10.144), P = 0.031)), and frequency of sexual intercourse ≥3 (COR = 5.793, 95%CI: (2.420, 13.869), P<0.001) were associated with UTI. In multivariate analysis, frequency of sexual intercourse (AOR = 7.907, 95% CI: (2.918, 21.425), P<0.001) and genital area washing habit (during defecation) AOR = 5.914, 95%CI: (1.860, 18.809), P = 0.003) and every morning AOR = 6.128, 95%CI: (1.602, 23.449), P = 0.008) were found to have statistically significant association with UTI. Out of a total of 94 female students who had significant uropathogen, 13 (13.8%) had ≥ 3 frequency of sexual activity (P<0.001). Furthermore, 45 (47.9%) participants indicated a habit of washing their genital area after defecation (P = 0.003), while 13 (13.8%) reported doing so every morning (P = 0.008). (Table 2).

**Table 2. Bivariate and Multivariate logistic regression analysis of risk factors for acquisition of asymptomatic UTI among female college students in Dessie town, Northeast Ethiopia, January 2021–March 2021.**

| Variable | Significant uropathogens | | COR (95% CI) | P-Value | AOR (95% CI) | P-Value |
|---|---|---|---|---|---|---|
| | No No. % | Yes No.% | | | | |
| **Age (in years)** | | | | | | |
| ≤20 | 136 (79.1) | 36 (20.9) | 1 | | | |
| 21–25 | 162 (77.5) | 47 (22.5) | 1.096 (0.671, 1.790) | 0.714 | NA | |
| 26–30 | 18 (72.0) | 7 (28.0) | 1.469 (0.570, 3.788) | 0.426 | NA | |
| >30 | 12 (75.0) | 7 (28.0) | 1.259 (0.383, 4.138) | 0.704 | NA | |
| **Residence** | | | | | | |
| Urban | 305 (78.6) | 83 (21.4) | | | | |
| Rural | 23 (67.6) | 11 (32.4) | 1.757 (0.823, 3.752) | 0.145 | 2.205 (0.904, 5.379) | 0.424 |
| **Student Batch** | | | | | | |
| Second year | 136 (76.8) | 41 (23.2) | 1 | | | |
| Third year | 129 (80.1) | 32 (19.9) | 0.823 (0.489, 1.386) | 0.463 | NA | |
| Fourth Year | 63 (75.0) | 21 (25.0) | 1.106 (0.604, 2.024 | 0.745 | NA | |
| **Marital Status** | | | | | | |
| Single | 256 (76.4) | 79 (23.6) | 1 | | | |
| Married | 57 (82.6) | 12 (17.4) | 0.682 (0.349, 1.335) | 0.264 | NA | |
| **Divorced** | 15 (83.3) | 3 (16.7) | 0.648 (0.183, 2.296) | 0.502 | NA | |
| **Family Monthly Income (ETB)** | | | | | | |
| < 1000 | 26 (70.3) | 11 (29.7) | 1.437 (0.666, 3.099) | 0.355 | NA | |
| 1001–2000 | 58 (80.6) | 14 (19.4) | 0.820 (0.424, 1.585) | 0.555 | NA | |
| 2001–3000 | 23 (82.1) | 5 (17.9) | 0.738 (0.268, 2.036) | 0.558 | NA | |
| 3001–4000 | 41 (78.8) | 11 (21.2) | 0.911 (0.438, 1.896) | 0.803 | NA | |
| >4000 | 180 (77.3) | 53 (22.7) | 1 | | | |
| **History of Catheterization** | | | | | | |
| Yes | 12 (63.2) | 7 (36.8) | 2.119 (0.810, 5.544) | 0.126 | 1.585 (0.512, 4.906) | 0.424 |
| No | 316 (78.4) | 87 (21.6) | 1 | | | |
| **DM status** | | | | | | |
| Yes | 8 (88.9) | 1 (11.1) | 0.430 (0.053, 3.483) | 0.429 | NA | |
| No | 320 (77.5) | 93 (22.5) | 1 | | | |
| **HIV /AIDS status** | | | | | | |
| Yes | 7 (77.8) | 2 (22.2) | 0.997 (0.204, 4.881) | 0.997 | NA | |
| No | 321 (77.7) | 92 (22.3) | 1 | | | |
| **History of STD** | | | | | | |

(*Continued*)

**Table 2.** (Continued)

| Variable | Significant uropathogens | | COR (95% CI) | P-Value | AOR (95% CI) | P-Value |
|---|---|---|---|---|---|---|
| | No No. % | Yes No.% | | | | |
| Yes | 2 (50) | 2 (50) | 3.543 (0.492, 25.500) | 0.209 | 4.546 (0.383, 53.975) | 0.230 |
| No | 326 (78) | 92 (22) | 1 | | | |
| **History of Hospitalization (last 3 month)** | | | | | | |
| Yes | 16 (66.7) | 8 (33.3) | 1.814 (0.751, 4.380) | 0.186 | 0.788 (0.165, 3.769) | 0.765 |
| No | 312 (78.4) | 86 (21.6) | 1 | | | |
| **History of Antibiotics use (last 3 month)** | | | | | | |
| Yes | 29 (78.7) | 13 (31.0) | 1.655 (0.823, 3.328) | 0.158 | 2.033 (0.614, 6.727) | 0.245 |
| No | 299 (78.7) | 81 (21.3) | 1 | | | |
| **Use of contraceptive** | | | | | | |
| IUD | 11 (91.7) | 1 (8.3) | 0.429 (0.054, 3.417) | 0.424 | 0.500 (0.057, 4.395) | 0.532 |
| Implant | 35 (68.6) | 16 (31.4) | 2.158 (1.097, 4.246) | 0.026 | 1.542 (0.676, 3.519) | 0.303 |
| Injection | 29 (67.4) | 14 (32.6) | 2.279 (1.112, 4.672) | 0.024 | 1.828 (0.794, 4.210) | 0.156 |
| Pill | 30 (73.2) | 11 (26.8) | 1.731 (0.805, 3.721) | 0.160 | 1.694 (0.684, 4.194) | 0.254 |
| Condom | 20 (69.0) | 9 (31.0) | 2.124 (0.906, 4.984) | 0.083 | 1.730 (0.664, 4.507) | 0.262 |
| No | 203 (82.5) | 43 (17.5) | 1 | | | |
| **Genitourinary abnormality** | | | | | | |
| Yes | 25 (69.4) | 11 (30.6) | 1.606 (0.759, 3.399) | 0.215 | 2.158 (0.843, 5.524) | 0.109 |
| No | 303 (78.5) | 83 (21.5) | 1 | | | |
| **Genital area washing habit** | | | | | | |
| During defecation | 54 (54.5) | 45 (45.5) | 3.452 (1.382, 8.622) | 0.008 | **5.914 (1.860, 18.809)** | **0.003***|
| During urination | 229 (88.8) | 29 (11.2) | 0.525 (0.211, 1.305) | 0.165 | 0.856 (0.274, 2.677) | 0.790 |
| Every morning | 16 (55.2) | 13 (44.8) | 3.366 (1.117, 10.144) | 0.031 | **6.128 (1.602, 23.449)** | **0.008***|
| Others | 29 (80.6) | 7 (19.4) | 1 | | | |
| **Frequency of sexual intercourse** | | | | | | |
| No | 254 (81.7) | 57 (18.3) | 1 | | | |
| <3 | 64 (72.7) | 24 (27.3) | 1.671 (0.964, 2.897) | 0.067 | 1.243 (0.624, 2.479) | 0.536 |
| > = 3 | 10 (43.5) | 13 (56.5) | 5.793 (2.420, 13.869) | 0.000 | **7.907 (2.918, 21.425)** | **0.000***|

Note:

*Statistically significant at P<0.05.

Others: under genital area washing habit include the females who wash their vaginal area 5–7 times per day.

1 = reference group, NA = not applicable.

## Bacterial and fungal uropathogen isolates

A total of 13 bacteria and 4 candida species were isolated from mid-stream urine sample. Almost all 103 (99.0%) of the infected female students had single infection; while only one (0.96%) participant had mixed infection (*C. albicans* and *C. krusei*), which makes the total number of bacterial isolates 57 and fungal isolates 47 (Table 3). The predominantly isolated uropathogens were *S. saprophyticus* 24 (23.07%) followed by *Candida tropicalis* 23 (22.1%), *Candida albicans* 10 (9.61%), *Candida krusei* 9 (8.65%) and *E. coli* 8 (7.69%).

## Antimicrobial susceptibility pattern of bacterial uropathogens

Majority of the isolated Gram negative uropathogens showed resistance for amoxicillin-clavulanic acid 92.3%. Rates of resistance of Gram-negative bacteria against tetracycline,

**Table 3. Distribution of bacterial and fungal uropathogens of asymptomatic UTI among female college students (N = 104) in Dessie town, Northeast Ethiopia, from January 2021 to March 2021.**

| Uropathogen isolates | N (%) |
|---|---|
| **Bacterial isolates** | |
| **Gram positive** | |
| *Staphylococcus saprophyticus* | 24 (23.07) |
| *Staphylococcus aureus* | 4 (3.84) |
| *Staphylococcus epidermidis* | 1 (0.96) |
| **Gram negative** | |
| *Escherichia coli* | 8 (7.69) |
| *Klebsiella ozenae* | 5 (1.92) |
| *Klebsiella rhinoscleromatis* | 3 (2.88) |
| *Providencia stuartii* | 3 (2.88) |
| *Citrobacter species* | 2 (3.5) |
| *Serratia species* | 2 (1.92) |
| *Citrobacter diversus* | 2 (1.92) |
| *Acinetobacter species* | 1 (0.96) |
| *Klebsiella oxytoca* | 1 (0.96) |
| *Klebsiella pneumonia* | 1 (0.96) |
| **Sub total** | **57 (54.8)** |
| **Fungi isolates** | |
| *Candida tropicalis* | 23 (22.11) |
| *Candida albicans* | 10 (9.61) |
| *Candida krusei* | 9 (8.65) |
| *Candida galberta* | 6 (5.76) |
| **Sub-total** | **47 (45.19)** |
| **Total** | **104 (100)** |

trimethoprim-sulfamethoxazole, ampicillin ranges from 52%–71.4%. However, all Gram-negative bacterial isolates showed relatively low level of resistance against norfloxacin 7.1%, gentamicin 8% and ciprofloxacin 10.7% (Table 4). Among the Gram-negative bacteria, all isolates showed high level of resistance to amoxicillin-clavulanic acid 100%, except *E. coli* 6 (75%). However, isolates of *E. coli* were highly sensitive to nitrofurantoin 100%, followed by norfloxacin 7 (87.5%) and gentamicin 6 (75%). *Klebsiella rihnoscleromatis* on the other hand were fully resistant 3 (100%) to trimethoprim-sulfamethoxazole, amoxicillin-clavulanic acid, ceftriaxone and tetracycline, but they were 3 (100%) sensitive to ciprofloxacin, nitrofurantoin, gentamicin and amikacin (Table 4).

Gram-positive uropathogens showed high level of resistance for penicillin 28 (96.6%) and trimethoprim-sulfamethoxazole 23 (79.3%). On the other hand, all gram-positive bacterial isolates showed higher sensitivity to nitrofurantoin 27 (93.1%), norfloxacin, clindamycin and ciprofloxacin 24 (82.7%), 20 (69%), 19 (65.5%) respectively (Table 5). *Staphylococci saprophyticus*, which were the predominant isolates among the gram positives, were resistant to penicillin and trimethoprim-sulfamethoxazole showing a resistant pattern of 22 (91.7%) and 20 (83.3%), respectively. Whereas nitrofurantoin and norfloxacin were found to be effective against 22 (91.7%) and 20 (83.3%) of *S. saprophyticus*, respectively. Likewise, *Staphylococcus aureus* showed resistance to penicillin 4 (100%), trimethoprim-sulfamethoxazole 3 (75%), and sensitivity to chloramphenicol 4 (100%), nitrofurantoin 4 (100%) and norfloxacin 3 (75%). *Staphylococcus epidermidis*, which were the least common isolate among gram positives, were fully

**Table 4. Antimicrobial susceptibility pattern of Gram-negative bacteria (n = 28) among asymptomatic female college students at Dessie town, Northeast Ethiopia, from January 2021 to March 2021.**

| Isolates (n) | Pattern | Antimicrobial agents | | | | | | | | | | | |
|---|---|---|---|---|---|---|---|---|---|---|---|---|---|
| | | CIP | TTC | SXT | F | NOR | CRO | AMC | CTX | CAZ | AMP | AMK | GN |
| E. coli (8) | S | 5 (62.5) | 2 (25.0) | 4 (50.0) | 8 (100.0) | 7 (87.5) | 5(62.5) | 0 (0.0) | 5(62.5) | 5 (62.5) | 0 (0.0) | 4 (50.0) | 6 (75.0) |
| | I | 1 (12.5) | 0 (0.0) | 1 (12.5) | 0 (0.0) | 0 (0.0) | 2 (25.0) | 2 (25.0) | 2 (25.0) | 1 (12.5) | 1 (12.5) | 1 (12.5) | 1 (12.5) |
| | R | 2 (25.0) | 6 (75.0) | 3 (37.5) | 0 (0.0) | 1 (12.5) | 1 (12.5) | 6 (75.0) | 1 (12.5) | 2 (25.0) | 7 (87.5) | 3 (37.5) | 1 (12.5) |
| K. ozenae (5) | S | 5 (100.0) | 2 (40.0) | 1(100.0) | 1 (20.0) | 5 (100.0) | 2 (40.0) | 0 (0.0) | 2 (40.0) | 1 (20.0) | | 4 (80.0) | 4 (80.0) |
| | I | 0 (0.0) | 3 (60.0) | 0 (0.0) | 0 (0.0) | 0 (0.0) | 0 (0.0) | (0.0) | 0 (0.0) | 1 (20.0) | NT | 0 (0.0) | 0 (0.0) |
| | R | 0 (0.0) | 0 (0.0) | 4 (80.0) | 4 (80.0) | 0 (0.0) | 3 (60.0) | 5 (100.0) | 3 (60.0) | 3 (60.0) | | 1 (20.0) | 1(20.0) |
| K. rhinoscleromatis (3) | S | 3 (100.0) | 0 (0.0) | 0 (0.0) | 3 (100.0) | 3 (100.0) | 0 (0.0) | 0 (0.0) | 0 (0.0) | 1 (33.3) | | 3 (100.0) | 3 (100.0) |
| | I | 0 (0.0) | 0 (0.0) | 0 (0.0) | 0 (0.0) | 0 (0.0) | 0 (0.0) | 0 (0.0) | 0 (0.0) | 0 (0.0) | NT | 0 (0.0) | 0 (0.0) |
| | R | 0 (0.0) | 3 (100.0) | 3 (100.0) | 0 (0.0) | 0 (0.0) | 3 (100.0) | 3 (100.0) | 3 (100.0) | 2 (66.7) | | 0 (0.0) | 0 (0.0) |
| | S | 1 (100.0) | 0 (0.0) | 0 (0.0) | 0 (0.0) | 1 (100.0) | 0 (0.0) | 0 (0.0) | 0 (0.0) | 0 (0.0) | | 0 (0.0) | 1 (100.0) |
| | I | 0 (0.0) | 0 (0.0) | 0 (0.0) | 0 (0.0) | 0 (0.0) | 1 (100.0) | 0 (0.0) | 1 (100.0) | 1 (100.0) | NT | 1 (100.0) | 0 (0.0) |
| | R | 0 (0.0) | 1 (100.0) | 1 (100.0) | 1 (100.0) | 0 (0.0) | 0 (0.0) | 1 (100.0) | 0 (0.0) | 0 (0.0) | | 0 (0.0) | 0 (0.0) |
| K. pneumonia (1) | S | 1 (100.0) | 1 (100.0) | 1(100.0) | 0 (0.0) | 0 (0.0) | 0 (0.0) | 0 (0.0) | 0 (0.0) | 1 (100.0) | | 1(100.0) | 1 (100.0) |
| | I | 0 (0.0) | 0 (0.0) | 0 (0.0) | 1 (100.0) | 1 (100.0) | 0 (0.0) | 0 (0.0) | 0 (0.0) | 0 (0.0) | NT | 0 (0.0) | 0 (0.0) |
| | R | 0 (0.0) | 0 (0.0) | 0 (0.0) | 0 (0.0) | 0 (0.0) | 1 (100.0) | 1 (100.0) | 1 (100.0) | 0 (0.0) | | 0 (0.0) | 0 (0.0) |
| C. diversus (2) | S | 2 (100.0) | 1 (50.0) | 2 (100.0) | 1 (50.0) | 2 (100.0) | 2 (100.0) | 0 (0.0) | 2 (100.0) | 2 (100.0) | 1 (50.0) | 2(100.0) | 1 (50.0) |
| | I | 0 (0.0) | 0 (0.0) | 0 (0.0) | 0 (0.0) | 0 (0.0) | 0 (0.0) | (0.0) | 0 (0.0) | 0 (0.0) | 0 (0.0) | 0 (0.0) | 1 (50.0) |
| | R | 0 (0.0) | 1 (50.0) | 0 (0.0) | 1 (50.0) | 0 (0.0) | 0 (0.0) | 2 (100.0) | 0 (0.0) | 0 (0.0) | 1(50.0) | 0 (0.0) | 0 (0.0) |
| Citrobacter spp. (2) | S | 0 (0.0) | 1 (50.0) | 1 (50.0) | 2 (100.0) | 2 (100.0) | 2 (100.0) | 0 (0.0) | 2 (100.0) | 0 (0.0) | 0 (0.0) | 2 (100.0) | 1 (50.0) |
| | I | 2 (100.0) | 0 (0.0) | 0 (0.0) | 0 (0.0) | 0 (0.0) | 0(0.0) | 0 (0.0) | 0(0.0) | 1 (50.0) | 1 (50.0) | 0 (0.0) | 1 (50.0) |
| | R | 0 (0.0) | 1 (50.0) | 1 (50.0) | 0 (0.0) | 0 (0.0) | 0 (0.0) | 2 (100.0) | 0 (0.0) | 1 (50.0) | 1 (50.0) | 0 (0.0) | 0 (0.0) |
| P. stuartii (3) | S | 3 (100.0) | | 0 (0.0) | 3 (100.0) | 2 (66.7) | 0 (0.0) | 0 (0.0) | 0 (0.0) | 1 (33.3) | 2 (66.7) | 1 (33.3) | |
| | I | 0 (0.0) | NT | 0 (0.0) | 0 (0.0) | 0 (0.0) | 2 (66.6) | 0 (0.0) | 2 (66.6) | 1(33.3) | 0 (0.0) | 1 (33.3) | NT |
| | R | 0 (0.0) | | 3 (100.0) | 0 (0.0) | 1 (33.3) | 1 (33.4) | 3 (100.0) | 1 (33.4) | 1 (33.3) | 1 (33.3) | 1 (33.3) | |
| Acinetobacter spp (1) | S | 1 (100.0) | 0 (0.0) | 0 (0.0) | 1(100.0) | 1(100.0) | 0(0.0) | 0 (0.0) | 0(0.0) | 0 (0.0) | | 0 (0.0) | 1 (100.0) |
| | I | 0 (0.0) | 0 (0.0) | 0 (0.0) | 0 (0.0) | 0 (0.0) | 0 (0.0) | 0 (0.0) | 0 (0.0) | 0 (0.0) | NT | 0(0.0) | 0 (0.0) |
| | R | 0 (0.0) | 1 (100.0) | 1(100.0) | 0 (0.0) | 0 (0.0) | 1(100.0) | 1 (100.0) | 1(100.0) | 1 (100.0) | | 1 (100.0) | 0 (0.0) |
| Serratia spp. (2) | S | 1 (50.0) | 1 (50.0) | 0 (0.0) | | 2 (100.0) | 1 (50.0) | | 1 (50.0) | 1 (50.0) | | 2 (100.0) | 2 (100.0) |
| | I | 0 (0.0) | 1 (50.0) | 2 (100.0) | NT | 0 (0.0) | 0 (0.0) | NT | 0 (0.0) | 0 (0.0) | NT | 0 (0.0) | 0 (0.0) |
| | R | 1 (50.0) | 0 (0.0) | 0 (0.0) | | 0 (0.0) | 1 (50.0) | | 1 (50.0) | 1 (50.0) | | 0 (0.0) | 0 (0.0) |
| Total (28) | S | 22 (78.5) | 8 (32.0) | 9(32.1) | 19 (73.0) | 25 (89.2) | 12 (42.8) | 0 (0.0) | 12 (42.8) | 12 (42.8) | 3 (21.4) | 19 (67.9) | 20 (80.0) |
| | I | 3 (10.7) | 4 (16.0) | 3 (10.71) | 1 (3.84) | 1 (3.57) | 5 (17.8) | 2 (7.7) | 5 (17.8) | 5 (17.8) | 1 (7.14) | 3 (10.7) | 3 (12.0) |
| | R | 3 (10.7) | 13 (52.0) | 16(57.1) | 6 (23.0) | 2 (7.14) | 11 (39.2) | 24 (92.3) | 11 (39.2) | 11 (39.2) | 10 (71.4) | 6 (21.4) | 2 (8.0) |

**CIP** = Ciprofloxacin, **TTC** = Tetracycline, **SXT** = Trimethoprim-Sulfamethoxazole, **F** = Nitrofurantoin, **NOR** = Norfloxacin, **CRO** = Ceftriaxone, **AMC** = Amoxacillin-Clavulinic acid, **CTX** = Cefotaxime, **CAZ** = Ceftazidime, **AMP** = Ampicillin, **AMK** = Amikacin, **GN** = Gentamicin, **R** = Resistant, **I** = Intermediate, **S** = Sensitive, **NT** = Not tested

resistant to penicillin and sensitive to chloramphenicol, nitrofurantoin and norfloxacin (Table 5).

## Multiple drug resistance patterns of bacterial isolates

Overall, all bacterial isolates were resistant to at least two antimicrobial agents whereas 54 (94.7%) isolates were resistant to ≥3 antimicrobial agents. Multidrug resistance (defined as

**Table 5. Antimicrobial susceptibility pattern of Gram-positive bacteria (n = 29) among asymptomatic female college students at Dessie town, Northeast Ethiopia, from January 2021 to March 2021.**

| Isolates (n) | Pattern | Antimicrobial agents | | | | | | | |
|---|---|---|---|---|---|---|---|---|---|
| | | CL | PEN | CAF | CIP | TTC | SXT | F | NOR |
| *S. aureus* (4) | S | 3 (75.0) | 0 (0.0) | 4(100) | 3 (75.0) | 2 (50.0) | 1 (25.0) | 4 (100.0) | 3 (75.0) |
| | I | 1 (25.0) | 0 (0.0) | 0 (0.0) | 1 (25.0) | 1 (25.0) | 0 (0.0) | 0 (0.0) | 1 (25.0) |
| | R | 0 (0.0) | 4 (100.0) | 0 (0.0) | 0 (0.0) | 1 (25.0) | 3 (75.0) | 0 (0.0) | 0 (0.0) |
| *S. epidermidis* (1) | S | 1 (100.0) | 0(0.0) | 1(100.0) | 1 (100.0) | 0 (0.0) | 0 (0.0) | 1(100.0) | 1 (100.0) |
| | I | 0 (0.0) | 0 (0.0) | 0 (0.0) | 0 (0.0) | 0 (0.0) | 1 (100.0) | 0 (0.0) | 0 (0.0) |
| | R | 0 (0.0) | 1 (100.0) | 0 (0.0) | 0 (0.0) | 1 (100.0) | 0 (0.0) | 0 (0.0) | 0 (0.0) |
| *S. saprophyticus* (24) | S | 16 (66.7) | 1 (4.2) | 9(37.5) | 15 (62.5) | 7 (29.2) | 3 (12.5) | 22 (91.7) | 20 (83.3) |
| | I | 4 (16.7) | 0 (0.0) | 0 (0.0) | 5 (20.8) | 5 (20.8) | 1 (4.2) | 2 (8.3) | 0 (0.0) |
| | R | 4 (16.7) | 22 (91.7) | 15 (62.5) | 4 (16.7) | 12 (50.0) | 20 (83.3) | 0 (0.0) | 4 (16.7) |
| Total (29) | S | 20 (69.0) | 1 (3.4) | 14(48.3) | 19 (65.5) | 9 (31.0) | 4 (13.7) | 27 (93.1) | 24 (82.7) |
| | I | 5 (17.2) | 0 (0.0) | 0 (0.0) | 6 (20.6) | 6 (20.6) | 2 (6.8) | 2 (6.8) | 1 (3.4) |
| | R | 4 (13.8) | 28 (96.6) | 15 (51.7) | 4 (13.7) | 14(48.2) | 23 (79.3) | 0 (0.0) | 4 (13.8) |

**CL** = Clindamycin, **E** = Erythromycin, **PEN** = Penicillin, **CAF** = Chloramphenicol, **CIP** = Ciprofloxacin, **TTC** = Tetracycline, **SXT** = Trimethoprim-Sulfamethoxazole, **F** = Nitrofurantoin, **NOR** = Norfloxacin, **R** = Resistant, **I** = Intermediate, **S** = Sensitive, **NT** = Not tested

non-susceptibility to 1 or more antimicrobial agent in 3 or more antimicrobial categories) was seen in 50 (87.7%) of all isolated bacterial uropathogens. With varying numbers, all types isolated bacterial species were found to be multidrug resistant. Collectively, 50 (85.7%) of Gram negatives and 26 (89.6%) of gram-positive bacteria showed multidrug resistance for the tested antimicrobial drugs (Table 6).

## Discussion

This research work was conducted to address the prevalence of asymptomatic UTI attributable to bacterial and fungal agents among female college students at Dessie town and it is also conducted to provide good information concerning antibiogram of bacterial isolates and the associated risk factors of the problem. After completing the study, we have obtained a result that indicates asymptomatic UTI is still public health concern and we have outlined some important clinical and associated factors of asymptomatic UTI for female college students. The overall prevalence of UTI was 24.6% (95% CI: 18.5, 26.1). Our finding is similar with a report in Addis Ababa 23.2% [31]. However, higher prevalence were reported from India 40% [35] and Bangladesh 62.6% [36], which might be due to variations in study participants' level of infection prevention and immunity status. These variations might be attributed to the fact that these studies were done on DM patients and hospitalized patients that have lower immunity status and strongly associated with acquisition of asymptomatic pathogens.

In our findings, the overall prevalence of bacterial UTI was 13.5% (95% CI, 10.2–16.6). This is in agreement with studies conducted among female students in Nigeria, 13.8% and 12% [37, 38]. Our finding is higher than the findings reported in Ghana, which reported an overall 9.6% prevalence of bacterial UTI among asymptomatic female students [16]. This discrepancy might be due to variation in the number of study participants; i.e., this study took a small number of asymptomatic female students due to high drop-out rate which might decrease the overall prevalence. However, our finding was lower than the study reported from Nigeria 46%–78% [39–41], which might be due to geographical variations. Similarly, higher report of asymptomatic bacteriuria was indicated in Ethiopia among diabetic patients (16.7%) [27],

**Table 6. Multi drug resistance patterns of bacterial isolates (n = 57) from asymptomatic female college students at Dessie town, Northeast Ethiopia, from January 2021 to March 2021.**

| Bacterial isolates | Antimicrobial resistance patterns | | | | | | Total | MDR |
|---|---|---|---|---|---|---|---|---|
| | R0 | R1 | R2 | R3 | R4 | ≥R5 | 20 (39.2) | |
| **Gram positives** | | | | | | | | |
| *S. aureus* | 0(0.0%) | 0(0.0%) | 1 (25.0%) | 2 (50.0%) | 1 (25.0%) | 0 (0.0%) | 4 (100.0%) | 3 (75.0%) |
| *S. epidermidis* | 0(0.0%) | 0(0.0%) | 0 (0.0%) | 1 (100.0%) | 0 (0.0%) | 0 (0.0%) | 1 (100.0%) | 1 (100.0%) |
| *S. saprophyticus.* | 0(0.0%) | 0(0.0%) | 2 (8.3%) | 2 (8.3%) | 11 (45.8%) | 9 (37.5%) | 24 (100%) | 22 (91.7%) |
| *Sub total* | 0(0.0%) | 0(0.0%) | 3 (10.3%) | 5 (17.2%) | 12 (41.3) | 9 (31.0) | 29 (100) | 26 (89.6) |
| **Gram negatives** | | | | | | | | |
| *E. coli* | 0(0.0%) | 0(0.0%) | 0 (0.0%) | 1 (12.5%) | 3 (37.5%) | 4 (50.0%) | 8 (100.0%) | 7 (87.5%) |
| *Citrobacter spp.* | 0(0.0%) | 0(0.0%) | 0 (0.0%) | 0 (0.0%) | 1 (50.0%) | 1 (50.0%) | 2 (100.0%) | 2 (100.0%) |
| *K. ozenae* | 0(0.0%) | 0(0.0%) | 0 (0.0%) | 1 (20.0%) | 1 (20.0%) | 3 (60.0%) | 5 (100.0%) | 4 (80.0%) |
| *P. stuartii* | 0(0.0%) | 0(0.0%) | 0 (0.0%) | 0 (0.0%) | 0 (0.0%) | 3 (100.0%) | 3 (100.0%) | 3 (100.0%) |
| *K. rhinoscleromatis* | 0(0.0%) | 0(0.0%) | 0 (0.0%) | 0 (0.0%) | 0 (0.0%) | 3 (100.0%) | 3 (100.0%) | 3 (100.0%) |
| *K. oxytoca* | 0(0.0%) | 0(0.0%) | 0 (0.0%) | 0 (0.0%) | 0 (0.0%) | 1 (100.0%) | 1 (100.0%) | 1 (100.0%) |
| *C. diversus* | 0(0.0%) | 1(50.0%) | 0 (0.0%) | 0 (0.0%) | 0 (0.0%) | 1 (50.0%) | 2 (100.0%) | 1 (50.0%) |
| *Acinetobacter spp* | 0(0.0%) | 0(0.0%) | 0 (0.0%) | 0 (0.0%) | 0 (0.0%) | 1 (100.0%) | 1 (100.0%) | 1 (100.0%) |
| *Serratia spp.* | 0(0.0%) | 0(0.0%) | 0 (0.0%) | 1 (50.0%) | 1 (50.0%) | 0 (0.0%) | 2 (100.0%) | 1 (50.0%) |
| *K. pneumoniae* | 0(0.0%) | 0(0.0%) | 0 (0.0%) | 0 (0.0%) | 0 (0.0%) | 1 (100.0%) | 1 (100.0%) | 1 (100.0%) |
| Sub-total | 0(0.0%) | 0(0.0%) | 0 (0.0%) | 3 (10.7%) | 6 (21.4%) | 18 (64.2%) | 28 (100.0%) | 24 (85.7) |
| **Total** | **0(0.0%)** | **1(1.7%)** | **3 (5.2%)** | **8 (14.0%)** | **18 (31.5%)** | **27 (47.3%)** | **57 (100.0%)** | **50 (87.7%)** |

R0 = no antibiotic resistance, **R1** = resistance to one, **R2** = Resistance to two, **R3** = Resistance to three, **R4** = Resistance to four, ≥ **R5** = Resistance to five and more drugs, **MDR** = multi-drug resistant

HIV patients (18%) [42], pediatric patients (15.9%) [29], which might be attributable to immune suppression of the DM and HIV patients that increases the susceptibility to bacterial infection. Furthermore, bacteriuria among pediatrics peak during infancy and toilet training period as a result of contamination [29].

Moreover, the prevalence of asymptomatic candiduria in this study was 11.1% (95% CI: 8.5–13.7), which is lower than a study done in Nigeria 40.7% [43]. Such variation might be attributed to difference in laboratory methods used to identify the microorganism and risk factors with geographical areas.

Unlike most of the studies reported in our country and elsewhere in the world, our findings showed majority of the etiological agents for Candiduria were *C. troipcalis* (48.9%), followed by *C. albicans* (21.2%). Previous studies reported that *C. albicans* was the predominant isolate (50–70%) [44]. This indicated that there is a paradigm shift toward non-albicans *Candida;* and it was demonstrated that the prevalence of *C. albicans* has significantly declined from 2013–2014 to 2015 [44]. The shift from *C. albicans* to *C. tropicalis* as the predominant Candiduria-causing species can be attributed to several factors. Firstly, antimicrobial resistance may have played a role. *Candida albicans*, being the most commonly isolated species in previous studies, may have developed resistance to commonly prescribed antifungal agents. This resistance can lead to reduced efficacy in treating *C. albicans* infections and create favorable conditions for the emergence of non-albicans Candida species, such as *C. tropicalis*. Secondly, changes in the host immune status, such as the prevalence of comorbidities or the use of immunosuppressive therapies, could have influenced the shift towards non-albicans Candida. *Candida tropicalis* may possess specific characteristics that allow it to evade or subvert the host immune response,

leading to its increased prevalence in Candiduria cases. Additionally, alterations in environmental conditions, such as temperature, humidity, or sanitation practices, may have also provided a more conducive environment for *C. tropicalis* to thrive and outcompete *C. albicans*.

In our findings, *S. saprophyticus* (42.1%) was the most predominant bacterial isolate followed by *E. coli* (14%). It is evident that *S. saprophyticus* causes UTI among sexually active young women due to displacement from the normal flora of the vagina and perineum into the urethra [32]. This finding is contrary with other studies, which reported *E. coli* as predominant bacterial isolate [27, 40, 42, 45]. However, *E. coli* was the most predominant isolate among gram negative bacterial isolates with an isolation rate of 8 (14.0%), which is supported by the previous studies conducted in Hawassa [42], Metu [27], Ghana [16], Nigeria [27, 37, 40, 42]. The key contributing factor for isolating such a high incidence of *E. coli* could be the presence of *E. coli* as a faecal flora, which then travels through the genitalia to induce UTI via contamination.; and due to numerous virulence factors used for colonization and invasion of the urinary epithelium such as P-fimbriae or pili adherence factors which mediate the attachment of *E. coli* to uroepithelial cells [32].

Our finding revealed that the frequency of sexual intercourse ($\geq$ 3 per week) showed significant association (P<0.001) with the prevalence of UTI. Females having a sexual habit $\geq$3 per week were 7.907 times more likely to acquire asymptomatic UTI than females that have a null sexual frequency habit, which is in agreement with studies in Nigeria [40, 41]. This could be due to the fact that female students within this age range (18–38 years) are sexually active with multi-sexual partners which may predispose them to UTI [32].

According to our investigation, prevalence of UTI in participants who had genital area washing habit after defecation (P = 0.003) and every morning (P = 0.008) was also 5.914- and 6.128 -times higher prevalence of UTI than their counterparts, respectively. This finding was consistent with similar studies reported from Nigeria [40, 46]. This could be as a result of poor genital hygienic practices mainly due to low frequency of genital washing and contamination of intestinal flora from feces to the urogenital area.

In our study, 10.5% of UTI occurred due to bacteria-fungi coinfection. This might be due to the fact that prior cases of urogenital infection predispose to fungal colonization since the earlier microbial colonizers might have altered the vaginal environment that creates conducive environment for Candida species to cause infection [43].

Mainly due to the habit of empirical treatment and infrequent bacterial identification and absence of susceptibility testing, antimicrobial resistance among bacterial uropathogens to the commonly used antibiotics become increasing that make clinicians left with very limited choices of drugs for the treatment of UTI [47]. In this study, the highest resistance was shown to penicillin (96.6%) and trimethoprim-sulfamethoxazole (79.3%) among gram positive bacteria. This could be due to the over use of these drug for many years. On the other hand, lower resistance (higher rate of sensitivity) was observed against nitrofurantoin, norfloxacin, clindamycin and ciprofloxacin. Similar findings have been reported in previous studies done in Dessie [45] and Nigeria [46]. The infrequency with which these medications are prescribed could be a possible explanation for such low-level resistance. As a result, they could be used as an alternative to antibiotics in the treatment of UTI.

Amoxicillin-clavulanic acid resistance was quite high in Gram negative bacteria (92.3%). This could be owing to the widespread availability and indiscriminate use of common medications like amoxicillin-clavulanic acid, which could contribute to an increase in resistance. On the contrary, all tested gram negative isolates showed sensitivity to nitrofurantoin (100.0%), norfloxacin (87.5%) and gentamicin (75.0%) which was also in agreement with the findings of other studies from Hawassa [42] and Nigeria [39, 46].

According to the international standard for definition of drug resistance [48], multidrug resistance (MDR = non-susceptible to ≥1 agent in ≥3 antimicrobial categories) was observed in 87.7% of bacterial isolates. This was higher than a study reported in Dessie 46.2% [45], Hawassa 78.3% [42] and Addis Ababa 81.1% [31]. This suggests that the rate of MDR was extremely high in routinely used antibiotics in the study area. The rise in MDR could be due to the overuse, and misuse of antimicrobial drugs, empirical way of treatment, poor infection prevention and control strategies, and transmission of resistant microorganisms in the health facilities.

The study had a limitation in that it could not include antifungal susceptibility testing due to the unavailability of antifungal agents in the market. However, it also had notable strengths. Firstly, it was the first study conducted in Ethiopia to assess the prevalence, antibiogram, and associated risk factors of asymptomatic uropathogens among college female students. Secondly, the study utilized CHROMagarTM Candida Plus media, which is the latest and pioneering chromogenic isolation medium, which is vital for the detection and differentiation of various Candida species, including *C. auris*, in addition to other significant species such as *C. albicans*, *C. tropicalis*, *C. glabrata*, and *C. krusei*.

## Conclusions

Both bacterial and fungi isolates were a major cause of asymptomatic UTI in sexually active female populations. In this study, a relatively similar prevalence of gram positive and gram-negative bacteria causing UTI was observed compared with other previous studies in the country. Nitrofurantoin and norfloxacin were effective for most of gram positive and gram-negative isolates whereas, penicillin, ampicillin, amoxicillin-clavulanic acid and trimethoprim-sulfamethoxazole were less effective for the management of UTI in the study area. Notably, increasing rate of resistance to the commonly used antimicrobial agents has been noticed for both gram negative and gram-positive bacterial isolates. Moreover, significant amount of multidrug resistance has been shown in most (87.7%) of the bacterial isolates. Frequency of sexual activity and genital area washing habit were significantly associated with acquisition of UTI among asymptomatic female college students. Hence, routine UTI screening, regular health education, reliance on AST-based treatment, and personal hygiene practices should be implemented to avoid the progression of an asymptomatic infection into a symptomatic UTI. Furthermore, to mitigate the impact of MDR, it is recommended to implement comprehensive infection control measures, including strict adherence to hand hygiene protocols, appropriate use of antimicrobial agents, and continuous surveillance of MDR organisms.

## Supporting information

**S1 Dataset.**
(PDF)

## Acknowledgments

The author would like to acknowledge Samara University and Wollo University for providing laboratory space and facilities to conduct the experiments. All selected colleges and all study participants are acknowledged for their cooperation during sample collection.

## Author Contributions

**Conceptualization:** Berhanu Kebede Reda, Genet Molla, Alemu Gedefie, Daniel Gebretsadik, Mihret Tilahun, Melaku Ashagrie Belete, Agumas Shibabaw.

**Data curation:** Berhanu Kebede Reda, Genet Molla, Alemu Gedefie, Daniel Gebretsadik, Mihret Tilahun, Melaku Ashagrie Belete, Agumas Shibabaw.

**Formal analysis:** Berhanu Kebede Reda, Genet Molla.

**Investigation:** Berhanu Kebede Reda.

**Methodology:** Berhanu Kebede Reda.

**Project administration:** Berhanu Kebede Reda.

**Resources:** Berhanu Kebede Reda, Melaku Ashagrie Belete, Agumas Shibabaw.

**Software:** Berhanu Kebede Reda, Alemu Gedefie.

**Supervision:** Genet Molla, Alemu Gedefie, Daniel Gebretsadik, Mihret Tilahun, Agumas Shibabaw.

**Validation:** Berhanu Kebede Reda.

**Writing – original draft:** Berhanu Kebede Reda.

**Writing – review & editing:** Berhanu Kebede Reda, Melaku Ashagrie Belete, Agumas Shibabaw.

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
