## [Decision Letter · Decision Letter 0]

31 Jan 2023

PONE-D-22-26752Magnitude and antibiogram of uropathogens and associated risk factors among asymptomatic female college students in Dessie town, Northeast EthiopiaPLOS ONE

Dear Dr. Berhanu Kebede Reda,

Thank you for submitting your manuscript to PLOS ONE. After careful consideration, we feel that it has merit but does not fully meet PLOS ONE’s publication criteria as it currently stands. Therefore, we invite you to submit a revised version of the manuscript that addresses the points raised during the review process.

We look forward to receiving your revised manuscript.

Kind regards,

Awatif Abid Al-Judaibi, PhD

Academic Editor

PLOS ONE

and https://journals.plos.org/plosone/s/file?id=ba62/PLOSOne_formatting_sample_title_authors_affiliations.pdf.

3. Please include your tables as part of your main manuscript and remove the individual files. Please note that supplementary tables (should remain/ be uploaded) as separate ""supporting information"" files.

Reviewers' comments:

Reviewer's Responses to Questions

**Comments to the Author**

1. Is the manuscript technically sound, and do the data support the conclusions?

Reviewer #1: Yes

Reviewer #2: Partly

Reviewer #3: Yes

2. Has the statistical analysis been performed appropriately and rigorously? 

Reviewer #1: Yes

Reviewer #2: No

Reviewer #3: Yes

3. Have the authors made all data underlying the findings in their manuscript fully available?

Reviewer #1: Yes

Reviewer #2: Yes

Reviewer #3: Yes

4. Is the manuscript presented in an intelligible fashion and written in standard English?

Reviewer #1: No

Reviewer #2: No

Reviewer #3: Yes

5. Review Comments to the Author

Reviewer #1: I found the manuscript very interesting because of ignored population but need improvement regarding conceptualization, language used in it ( English, consult with native)

1. Get through the whole Abstract need improvement in language

2. In line 4-6 authors use old reference, the ref should be updated of recent couple of years (update references in the whole manuscript if possible especially in introduction and discussion).

3. The sentence used in ref 3 is not strong statement, rephrase it.

4. From line 27 to ref 14, statements are not clear having grammatical mistakes, kindly correct it and also use new refs.

5. In discussion portion, in line 9-14 language typos are found kindly correct it.

6. In line 17-20 (Conclusion portion) the future prospective is not clear.

Reviewer #2: Reviewer’s comments on Manuscript PONE-D-22-26752

Title: Magnitude and antibiogram of uropathogens and associated risk factors among asymptomatic female college students in Dessie town, Northeast Ethiopia

General Comment

I have provided my comments on the document for actions.

Title: I suggest the title - Antibiogram of Uropathogens and associated risk factors among asymptomatic female college students in Dessie town, Northeast Ethiopia. Delete “Magnitude and” Please feel free to amend.

Short title: Same thing Delete “Magnitude

ABSTRACT:

Line 1- Delete “Asymptomatic” in the bracket and replace with “asymptomatic”

Line 7- Delete “asymptomatic”. It is repeated twice in the sentence. The sentence can read “to determine the prevalence and factors associated with urinary tract infection, and antibiogram of the uropathogen isolates among asymptomatic female college students”

Line 9 – Delete “found”

Line 10- Do you have any listed criteria for inclusion and exclusion of participants?

Also, what do you mean by "reproductive age"? Can you explain by including or representing this with an age group or age range?

Lines 13-16- This can read as follows for clarity “Data were coded and entered for statistical analysis using SPSS version 22.0. Descriptive statistics, bivariate and multivariate logistic regression analysis were performed and p-values ≤0.05 with corresponding 95% confidence interval (CI) were considered statistically significant”

Line 19 - What does S mean? Provide the full name of pathogen before abbreviation.

Line 24- please put commas in between the drugs – “Gram positive bacterial isolates were sensitive to norfloxacin, clindamycin, and ciprofloxacin accounting for 24 (82.7%), 20 (69.0%), and 19 (65.5%), respectively.

Lines 28-32 – please these lines are unclear. Rephrase for clarity. You may want to start by saying “Factors identified to be associated with UTI in asymptomatic female college students included frequency of sexual intercourse -------------------------------“what do you mean by every morning? Early morning what?

Lines 34-34- regular health education on the risk of asymptomatic infectious diseases for who?

INTRODUCTION

Line 7- delete “in a community”

Line 11- add “to be” affected

Lines 14-15 – Please rephrase the sentence or delete “while in symptomatic, signs are the major one”.

Line 18 – add full stop at the end of the sentence,

Line 19 “delete “ASB” and stick to UTI

Line 23: delete “the most”

Page 2 lines 5-6 not clear- please rephrase

MATERIALS AND METODS

Line 17- from January 2021 - March 2021 should end the sentence

Line 18-19- how was this done? Please provide details”. Also provide details of how participants were systematically selected.

Please provide a section on how the sample size for this study was estimated

Provide your inclusion and exclusion criteria for selection of participants

Please provide a reference number for the ethical approval

Data and specimen collection

Line 27- provide details of how the questionnaire was designed and structured? Were they personally administered or not? What demographic, clinical, and risk factors data were collected? Are questions contingency, matrix, open or closed ended questions? Was this tool pretested? How?

Line 28- change to “10 mL”. Please effect changes to the unit throughout the document

Pg 3 Line 14- CFU/ml change to “CFU/mL”. Effect changes throughout the document

Pg 4 Line 2- standard clinical laboratory methods (how? please be detailed)

Line 24- ”as a resistant” delete

Line 29- What are these quality control tests performed. Please describe

Line 30- Questionnaire a section for questionnaire design and administration in the materials and methods preferably before specimen collection section. Delete everything on questionnaire on this pg and pg 5 and move to the new section for the questionnaire design and administration. How was the question design and based on what? How did you structure your questionnaire? Summary of the questions asked in each section and number of questions? How did you pretest? How did you check the reliability performance of the questionnaire?

Statistical analysis

This is not detailed enough for the statistical analyses methods tools used for your data. Your descriptive statistics was computed as what? What types of data were collected for this study? Provide comprehensive illustration of steps undertaken for your binary and multivariable logistic regression? What were the independent variables tested for association? What is your dependent? How did you deal with confounding factors? Stepwise illustration of your multivariable model analysis is required? What informed factors (from the univariable analysis) fitted into the model?

Ethical approval and consent – Provide reference number

Results

Pg 6 Line 8 – delete “around”

Line 11- include factors (with p values, COR, and 95% CI) that were associated with UTI at bivariate level before your results for the multivariate regression.

Pg 7- Line 3 Antimicrobial susceptibility pattern of bacterial uropathogens

Can you mention these bacteria and their AMR pattern/profile?

Multiple drug resistance patterns of the isolates

This whole section is not detailed enough and too vague.

Around 85.7% of Gram negative 5 and 89.6% of gram-positive bacteria showed multidrug resistance for the tested antimicrobial 6 drugs (which gram negative bacteria; which gram positive bacteria; to which AMs drugs)

Discussion

First paragraph- please can you rephrase? There are some grammatical errors making your thoughts difficult to follow. Please, you will need to rewrite this section. A few of your arguments are really not logical. There are grammatical and typo errors making this hard to follow

Conclusion

Where participants being managed for UTI?

Figure is not clear. I feel this should be deleted, looks repetitive since you have mentioned in the text (the result section)

Reviewer #3: Manuscript title: Magnitude and antibiogram of uropathogens and associated risk factors among asymptomatic female college students in Dessie town, Northeast Ethiopia

Thank you, the Editor PLOS ONE Global Public Health, for the invite to review the above-named manuscript. The authors research topic is public health significance with the perspective of the laboratory component of the study.

Please see below comments for the kind consideration of the authors aimed at adding value to the research work. Thank you.

General comment

1. The line numbering was not consistent across the body of the manuscript text. Authors need to ensure proper line numbering.

Abstract

1. Objectives: The objectives are captured as if the study is yet to be done. Authors need to frame the objectives in past tense.

2. Methods: p value supposed to be <0.05 and not ≤ 0.05

3. Result: Authors should maintain 1 decimal place for the proportion, and 2 decimal places for the Adjusted odds ratio and 95% confidence interval. P-value should also be in 2 decimal places.

Materials and methods

Line number 15: Suggested sequence to follow as follows: Study area, Study design, Study population, Eligibility criteria (Inclusion and Exclusion criteria), Sample size determination, Sampling technique, Data collection method(s) and Data analysis.

Line numbers 17-25: Need for authors to reframe the methods section.

Line numbers 19-22: Authors need to provide detailed information on how the study areas were selected.

Line number 23: This is the sample size, authors need to provide detailed information on how the sample size calculation was done.

Line number 22: ‘’female college students’’: These are study population. There is need for authors to incorporate eligibility criteria (inclusion and exclusion criteria)

Line numbers 23-25: Authors should provide detailed information on the sampling technique used to select eligible respondents for interview

Line number 27: Authors to state the method of questionnaire administration: self or interviewer administered?

Line number 28: Clinical and risk factors data on ………..? Need for the authors to provide the need information.

Line number 5 under specimen transportation: ‘’status’’ to be added after non-pregnancy

Line 22 under Cultivation and identification of isolates: was changed to ‘’were’’

Line 28 under Cultivation and identification of isolates: sub cultured changed to subcultured

Lines 13-20 under Antimicrobial Susceptibility: Drug names to start with Capital letter

Line 25 under Antimicrobial Susceptibility: discs ‘’was’’ measured ……. Change ‘’was’’ to ‘’were’’

Line 26 under Antimicrobial Susceptibility: susceptibility tests ‘’was’’ based …… Change ‘’was’’ to ‘’were’’

Line 3 under Quality assurance: questionnaires ‘’was’’ checked by … Change ‘’was’’ to ‘’were’’

Lines 3-5: ‘’The questionnaire was pretested on 5%..........., Northeast Ethiopia. It will be good if the authors could provide information on feedback from the pre-test and it helped on improving the quality of the study.

Lines 21-22: ‘’To show the relationship between each variable……was utilized’’. Test statistic to assess association between dependent and independent variables is Odds ratio (Crude OR) (with 95% CI) OR Chi square (with p-value). This is bivariate analysis. Binary logistic regression is a multivariate analysis used to assess predictors of a dependent or outcome variable. Authors should also provide information on Dependent (outcome) and Independent (explanatory) variables. The sequence supposed to be univariate, bivariate, and multivariate analysis.

Line 25: Ethical approval and consent to participate: For journal publication, authors should situate the ethical considerations sub-section based on the journal's guidelines.

Results

Line 5: Authors should include the response rate based on the number of eligible respondents reached.

Line 9: Table 1 to be included after students

Line number 14 ‘’15 independent variables’’; Authors should provide information on the independent variables under the methods section

Authors should maintain 2 decimal points for information on results section.

Line 16: AOR: Adjusted Odds Ratio. This should be mentioned under the methods section and written in full with the binary logistic regression done.

Line 20: Authors should use greater than or equal to sign

Line 21: (47.9) and (13.8), Authors to insert % to 47.9 and 13.8

Line 3 under Multiple drug resistance patterns of the isolates: This should be defined under the methods section. Need for authors to include ''Operational definitions'' under the methods section.

Discussion

The first paragraph of the Discussion should first provide information on summary of the study based on the study objectives. This is important to provide information if the study objectives is achieved or not.

Line 11 and 18: The use of the ''i.e'' acronym should be deleted.

Lines 15-16: ‘’This is in agreement with studies in Nigeria’’ This is incomplete. Need for authors to provide information on the type of study conducted.

Line 17: relatively higher than the findings in Ghana: Need to elaborate on the study information in Ghana.

Line 20: ‘’findings was lower….from Nigeria’’: Authors should be explicit on the type of information in relation to Nigeria.

Line 24 and 25: ‘’Furthermore, bacteriuria among pediatircs peak during ……………….’’ Reference to be provided.

Line 20-23: ‘’Our finding revealed that the frequency of sexual intercourse………: This is repetition of results, authors are advised to delete.

Lines 26-28: ‘’According to our investigation, prevalence of UTI……..’’ This is repetition of results, authors are advised to delete.

Line 8-11: ‘’ In the present study, there was no statistically significant association between prevalence of UTI in asymptomatic female college students and age, residence (colleges), student batch, monthly family income, marital status, history of UTI, history of catheterization, history of STDs, and presence of genitourinary abnormalities. Similar finding was reported from Nigeria’’. I do not think this statement is important.

Line 1: According to the international standard for definition of drug resistance ‘‘This information supposed to be part of the operational definitions under the methods section’’.

Limitations

Lines 22 and 23: Study limitations are factors which could affect the outcome of the study but however, measures were taken in addressing or mitigating such. The information provided by the authors is not a study limitation.

Line 24: List of abbreviations: Authors should ensure the list of abbreviations should be captured the way they appear in the body of the manuscript.

Figure 1: The title should be underneath the chart.

Table 1: Unequal class width for respondents age. Authors should ensure class width are equal. What does ETB means? This supposed to be captured under the abbreviation section.

Table 2: Authors should maintain 2 decimal points all through.

Others: under genital area washing habit include the females who wash their vaginal area 5-7 times per day. SUP

= Significant Uropathogen, AOR = adjusted odds ratio, COR = crude odds ratio, 1 = reference group, 95% CI = 95% confidence interval, DM = Diabetic mellitus, UTI = urinary tract infection, NA = not applicable. ‘’This is to be included in the abbreviation section.

Table 3: Frequency of bacterial and fungal uropathogens of asymptomatic UTI among female college students in Dessie town, Northeast Ethiopia, January 2021 - March 2021.: Authors to include N=104 with the title. ‘’from 104’’ to be deleted from N (%)

Antimicrobial resistance pattern (R0-R5) in Table 6: The antimicrobial resistance pattern supposed to be mentioned under the operational definitions of the methods section.

6. PLOS authors have the option to publish the peer review history of their article (what does this mean?). If published, this will include your full peer review and any attached files.

Reviewer #1: No

Reviewer #2: No

Reviewer #3: No

---

## [Author Response · Author response to Decision Letter 0]

3 Mar 2023

All the questions raised by the Reviewers have been addressed; and the manuscript is modified accordingly. Moreover, all the requested Editorial corrections are addressed in both the revised manuscript and the response letter. 

A point-by-point response to reviewer comments and queries are provided in the file labeled 'Response to reviewers'.

---

## [Editor Report · Decision Letter 1]

10 Mar 2023

PONE-D-22-26752R1

Antibiogram of uropathogens and associated risk factors among asymptomatic female college students in Dessie town, Northeast Ethiopia

PLOS ONE

Dear Dr. Reda,

Thank you very much for submitting your manuscript to PLOS ONE, and for responding to our recent requests regarding your submission. Unfortunately, in our final editorial checks of the documents that you supplied, we have concluded that your submission does not our ethical requirements for human subjects research submissions. We will therefore be overturning the provisional editorial accept decision, and will reject this manuscript.

PLOS ONE requires that research meets all applicable standards for the ethics of experimentation and research integrity (http://journals.plos.org/plosone/s/human-subjects-research). We reserve the right to reject any submission that does not meet our internal ethical standards, which in some cases are more stringent than local ethical standards.

In this case, the ethics approval document that you provided does not meet our standards for approval documentation. It is therefore not clear whether you obtained the necessary ethical approval for the study to take place.

As a result of this concern, we cannot consider the manuscript for publication. I am very sorry that this issue was identified at such a late stage.

Kind regards,

Emily Chenette

Editor in Chief

PLOS ONE
---

## [Author Response · Author response to Decision Letter 1]

20 Apr 2023

Detail responses to specific reviewer and editor comments are provided in the 'Response to reviewers' file.

---

## [Decision Letter · Decision Letter 2]

7 Jun 2023

PONE-D-22-26752R2

Antibiogram of uropathogens and associated risk factors among asymptomatic female college students in Dessie town, Northeast Ethiopia

PLOS ONE

Dear Dr. Berhanu Kebede Reda,

Thank you for submitting your manuscript to PLOS ONE. After careful consideration, we feel that it has merit but does not fully meet PLOS ONE’s publication criteria as it currently stands. Therefore, we invite you to submit a revised version of the manuscript that addresses the points raised during the review process.

We look forward to receiving your revised manuscript.

Kind regards,

Awatif Abid Al-Judaibi, PhD

Academic Editor

PLOS ONE

Journal Requirements:

Reviewers' comments:

Reviewer's Responses to Questions

**Comments to the Author**

1. If the authors have adequately addressed your comments raised in a previous round of review and you feel that this manuscript is now acceptable for publication, you may indicate that here to bypass the “Comments to the Author” section, enter your conflict of interest statement in the “Confidential to Editor” section, and submit your "Accept" recommendation.

Reviewer #2: All comments have been addressed

Reviewer #4: (No Response)

2. Is the manuscript technically sound, and do the data support the conclusions?

Reviewer #2: Yes

Reviewer #4: Partly

3. Has the statistical analysis been performed appropriately and rigorously? 

Reviewer #2: Yes

Reviewer #4: Yes

4. Have the authors made all data underlying the findings in their manuscript fully available?

Reviewer #2: Yes

Reviewer #4: Yes

5. Is the manuscript presented in an intelligible fashion and written in standard English?

Reviewer #2: No

Reviewer #4: No

6. Review Comments to the Author

Reviewer #2: Title: Antibiogram of uropathogens and associated risk factors among 3 asymptomatic female college students in Dessie town, Northeast Ethiopia

Revised Manuscript

General comment: The authors have tried to address my concerns. And the manuscript reads better and detailed. Minor corrections and recommend publication. However, there is still need for English language editing.

Many thanks

Abstract:

Background

Line 3- You need to open and close the bracket - (on asymptomatic bacteriuria and asymptomatic candiduria)

Methods

Line 13 – Ten mLs

Introduction

Lines 27-29- provide reference

Materials and methods

Talk about the study area before the design.

Antimicrobial susceptibility testing

Line 2- - change to 5mLs

Quality assurance

Line 12-rephrase to read “An hour training was given to data collectors on data collection”.

Lines 12-16 looks like a repetition “The questionnaire was pretested on 5%

of the sample size (22 asymptomatic female students) from Mankul College which is located

in Dessie city, Northeast Ethiopia, and necessary modifications were made to the data

collection tool accordingly, particularly to improve the clarity of questions”

Reviewer #4: Title: Antibiogram of uropathogens and associated risk factors among asymptomatic female college students in Dessie town, Northeast Ethiopia

Manuscript Number PONE-D-22-26752

I acknowledge the editor of PLOS ONE for inviting me to review this manuscript. I would also like to appreciate the authors for doing this research.

General comments:

The manuscript has been well written but still has errors and needs an improvement. There are grammatical and punctuation issues that need revision.

Abstract:

Line 13: “Ten mL”, need capitalization issue

Line 24: 24 (92.3), need and edition by adding “%”

Introduction

Line 2-6: is a full of grammatical errors and scientific justification is required. The expression “Asymptomatic uropathogens” should be revised and rewritten in the right way because infected parson, not the uropathogen will develop symptom or remain asymptomatic.

“Asymptomatic uropathogens” occur frequently and are “a” major cause of UTI leading to unnecessary treatments 5 [2]). It affects millions of people globally with possible conversion into severe symptomatic infections, “especially in developed countries where the healthcare coverage is comparably low.”

Page 5:

Line 11-21, need rearrangement and merging of related texts and put numerical data in different studies in ranges. But the unrelated groups”HIV” patients” should better be removed.

Line 26-29: the sentence is long complex with grammatical errors.

MATERIAL AND METHODS:

“Memhran College (MC)”

Page 7

Line 6: How could the variable “educational level” be important? All study participants and were college students.

Line 17-19 “…within two hours for processing after confirming their non-pregnancy status by using 19 human chorionic gonadotropin (HCG) card pregnancy test.” How much was the period of delay before HCG testing?

Line 24: “Cultures were incubated overnight” dear authors please revise this and replace “cultures” by inoculated media or …media or plates.

Results

Page 10: The median age of the study participants was 21 years, ranges from 18 – 38 years

Is 18–38 the interquartile range? Is 18 years the lowest range?

Page 11:

Line 4-5: “Majority” of the study 5 participants 209 (49.4%) were in the age group of 21 to 25 years. how less than half (49.4%) can be majority?

209 (49.4%)

Line 26-28: Probably the strength of this study is the sample size“…, this study took a small number of asymptomatic female students …”

Discussion:

Page 14:

Line 11-16: “Unlike most of the studies reported in our country and elsewhere in the world, our findings showed majority of the etiological agents for Candiduria were C. troipcalis (48.9%), followed by C. albicans (21.2%).”

Would the authors discuss adequately on why shift from C. albicans to C. troipcalis was seen?

Page 15:

Line 5-6: Age (“17”-30 years) are sexually active with multi-sexual partners? Which age group was your population?

Which age “15, 17 or 18” years the lowest range. On page 10: line 3-4, 18 it seems to be 18 years, Page 15, Line 5-6: “17” and in the table, it is mentioned as 15-20, …

Dear authors:

Is your research spotless? I haven’t see any limitation?

A high rate of MDR has been reported but there is no recommendation regarding this serious issue. There has to be recommendation about MDR (87.7%) infection.

7. PLOS authors have the option to publish the peer review history of their article (what does this mean?). If published, this will include your full peer review and any attached files.

Reviewer #2: No

Reviewer #4: No

---

## [Author Response · Author response to Decision Letter 2]

18 Jun 2023

A point-by-point response to reviewer and editor comments were fully provided in the file labeled "Response to Reviewers".

---

## [Editor Report · Decision Letter 3]

6 Jul 2023

Antibiogram of uropathogens and associated risk factors among asymptomatic female college students in Dessie town, Northeast Ethiopia

PONE-D-22-26752R3

Dear Dr. Berhanu Kebede Reda,

We’re pleased to inform you that your manuscript has been judged scientifically suitable for publication and will be formally accepted for publication once it meets all outstanding technical requirements.

Kind regards,

Awatif Abid Al-Judaibi, PhD

Academic Editor

PLOS ONE

---

## [Editor Report · Acceptance letter]

10 Jul 2023

PONE-D-22-26752R3 

Antibiogram of uropathogens and associated risk factors among asymptomatic female college students in Dessie town, Northeast Ethiopia 

Dear Dr. Reda:

I'm pleased to inform you that your manuscript has been deemed suitable for publication in PLOS ONE. Congratulations! Your manuscript is now with our production department. 

Kind regards, 

on behalf of

Professor Awatif Abid Al-Judaibi 

Academic Editor

PLOS ONE